# Functional roles of Mg$^{2+}$ binding sites in ion-dependent gating of a Mg$^{2+}$ channel, MgtE, revealed by solution NMR

Tatsuro Maruyama[1], Shunsuke Imai[1], Tsukasa Kusakizako[2], Motoyuki Hattori[3,4,5], Ryuichiro Ishitani[2], Osamu Nureki[2], Koichi Ito[6], Andrès D Maturana[7], Ichio Shimada[1]*, Masanori Osawa[1,8]*

[1]Department of Physical Chemistry, Graduate School of Pharmaceutical Sciences, The University of Tokyo, Tokyo, Japan; [2]Department of Biological Sciences, Graduate School of Science, The University of Tokyo, Tokyo, Japan; [3]State Key Laboratory of Genetic Engineering, School of Life Sciences, Fudan University, Shanghai, China; [4]Collaborative Innovation Center of Genetics and Development, School of Life Sciences, Fudan University, Shanghai, China; [5]Department of Physiology and Biophysics, School of Life Sciences, Fudan University, Shanghai, China; [6]Department of Computational Biology and Medical Sciences, Graduate School of Frontier Sciences, The University of Tokyo, Chiba, Japan; [7]Department of Bioengineering Sciences, Graduate School of Bioagricultural Sciences, Nagoya University, Nagoya, Japan; [8]Division of Physics for Life Functions, Faculty of Pharmacy, Keio University, Tokyo, Japan

*For correspondence:
shimada@iw-nmr.f.u-tokyo.ac.jp (IS);
osawa-ms@pha.keio.ac.jp (MO)

Competing interests: The authors declare that no competing interests exist.

**Abstract** Magnesium ions (Mg$^{2+}$) are divalent cations essential for various cellular functions. Mg$^{2+}$ homeostasis is maintained through Mg$^{2+}$ channels such as MgtE, a prokaryotic Mg$^{2+}$ channel whose gating is regulated by intracellular Mg$^{2+}$ levels. Our previous crystal structure of MgtE in the Mg$^{2+}$-bound, closed state revealed the existence of seven crystallographically-independent Mg$^{2+}$-binding sites, Mg1–Mg7. The role of Mg$^{2+}$-binding to each site in channel closure remains unknown. Here, we investigated Mg$^{2+}$-dependent changes in the structure and dynamics of MgtE using nuclear magnetic resonance spectroscopy. Mg$^{2+}$-titration experiments, using wild-type and mutant forms of MgtE, revealed that the Mg$^{2+}$ binding sites Mg1, Mg2, Mg3, and Mg6, exhibited cooperativity and a higher affinity for Mg$^{2+}$, enabling the remaining Mg$^{2+}$ binding sites, Mg4, Mg5, and Mg7, to play important roles in channel closure. This study revealed the role of each Mg$^{2+}$-binding site in MgtE gating, underlying the mechanism of cellular Mg$^{2+}$ homeostasis.
DOI: https://doi.org/10.7554/eLife.31596.001

## Introduction

Magnesium ions (Mg$^{2+}$) are the most abundant divalent metal ions within cells. They bind to a number of proteins and nucleic acids, regulating a wide range of biological processes such as ATP utilization, enzyme activation, and maintenance of genomic stability (*Hartwig, 2001*; *Cowan, 2002*). In humans, abnormal Mg$^{2+}$ homeostasis is reportedly associated with several diseases including cardiovascular disease, diabetes, and high blood pressure (*Alexander et al., 2008*).

Cellular Mg$^{2+}$ homeostasis is maintained by a class of transmembrane proteins termed Mg$^{2+}$ transporters. The MgtE family of Mg$^{2+}$ transporters in bacteria, which operates as Mg$^{2+}$ channels (*Hattori et al., 2009*), is homologous to the eukaryotic SLC41 family (*Goytain and Quamme, 2005a*, *2005b*; *Kolisek et al., 2008*; *Moomaw and Maguire, 2008*; *Sahni and Scharenberg, 2013*).

With respect to the role of the MgtE family in $Mg^{2+}$ homeostasis, a $Mg^{2+}$-sensor riboswitch upregulates the gene expression of MgtE in response to a decrease in intracellular $Mg^{2+}$ levels (*Dann et al., 2007*); MgtE then restores and maintains the intracellular $Mg^{2+}$ homeostasis. Our prior electrophysiological study of MgtE from *Thermus thermophilus* revealed that MgtE facilitates $Mg^{2+}$ uptake across the cell membrane at intracellular $Mg^{2+}$ concentrations lower than 5–10 mM, whereas at higher $Mg^{2+}$ concentrations, MgtE does not allow further $Mg^{2+}$ uptake (*Hattori et al., 2009*).

We have reported the crystal structure of full-length MgtE in the $Mg^{2+}$-bound state, which revealed that it forms a homodimer comprised of an N-terminal cytoplasmic (CP) region (residues 1–263) and a C-terminal transmembrane (TM) region (residues 264–450) (*Figure 1A*) (*Hattori et al., 2007b*). The CP region consists of three parts, an N domain (residues 1–131), a cystathionine-β-synthase (CBS) domain (residues 132–245), and a plug helix (residues 246–263); the plug helix connects the latter with the TM region. The two subunits in the dimer are related by a crystallographic 2-fold symmetry, with the ion-conducting pore being formed along the axis of symmetry at the centre of the dimer in the TM region. The ion-conducting pore is closed at the CP side of the TM region, which is stabilized by the interactions with the C-termini of the plug helices (*Figure 1—figure supplement 1*). Thus, this portion appears to act as a gate for $Mg^{2+}$ transport, with the structure obtained reflecting the closed MgtE state (*Hattori et al., 2007b*).

In the crystal structure, seven crystallographically independent $Mg^{2+}$-binding sites (Mg1–Mg7) can be identified, among which Mg1 is located on the 2-fold axis in the ion-conducting pore at the extracellular side of the TM region, whereas the other six sites are located in the CP region (*Figure 1B*). The $Mg^{2+}$ ions bound to these sites coordinate with acidic residues, through at least one carboxylic group, and further bridge two different MgtE domains, either in the same or other subunits. Recently, the structural basis for ion selectivity was revealed through a high resolution crystal structure of the TM region, in which the Mg1 site accommodates a hydrated $Mg^{2+}$

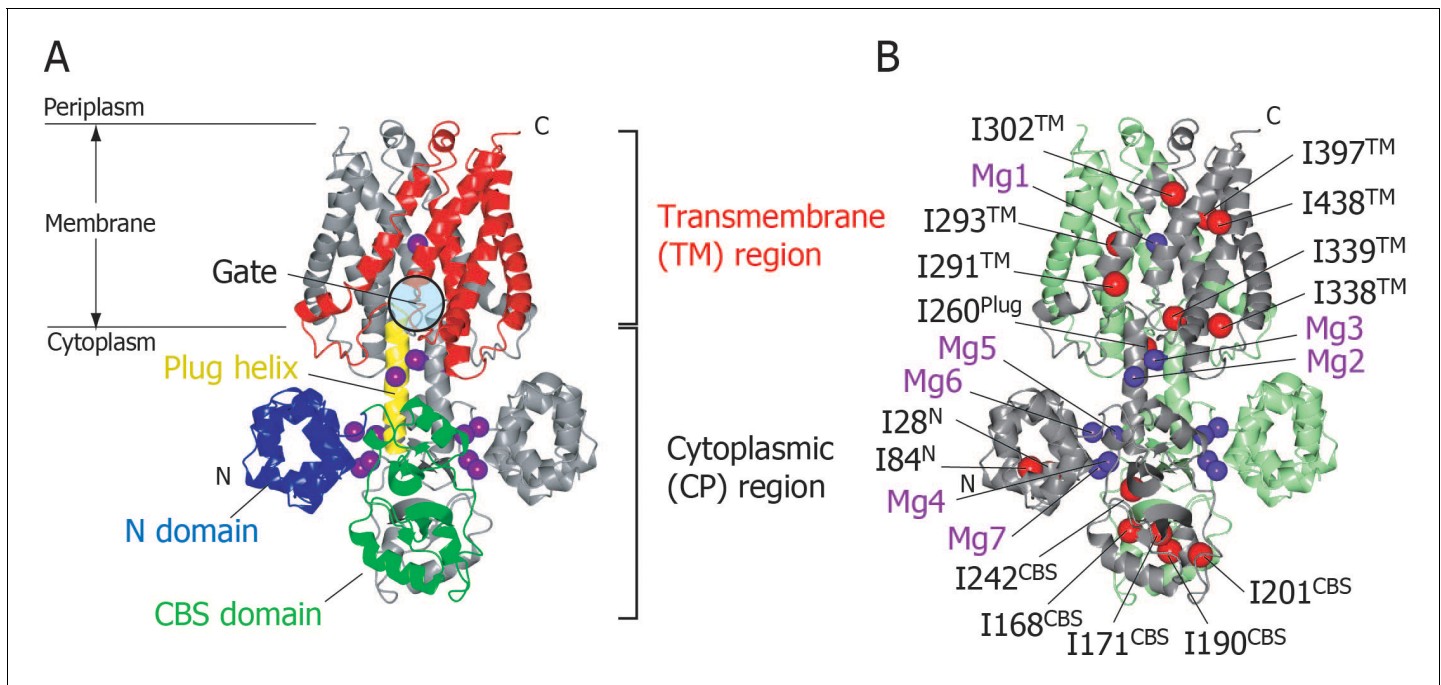

**Figure 1.** Structure of MgtE in the $Mg^{2+}$-bound state. (A) The MgtE dimer (PDB code:2ZY9) viewed in the membrane plane, with the N domain (blue), the CBS domain (green), the plug helix (yellow), and the TM region (red) highlighted in one subunit. $Mg^{2+}$ ions are shown as purple spheres. The gate region is circled (see *Figure 1—figure supplement 1*). (B) Two subunits of the dimer are shown in green and grey, respectively. Ile Cδ1 atoms in the grey subunit are shown as red spheres. The $Mg^{2+}$-binding sites in the dimer, Mg1-Mg7, are also labelled.

DOI: https://doi.org/10.7554/eLife.31596.002

The following figure supplement is available for figure 1:

**Figure supplement 1.** Gate region of MgtE.
DOI: https://doi.org/10.7554/eLife.31596.003

(*Takeda et al., 2014*). Electrophysiological investigation of MgtE $Mg^{2+}$-binding site mutants indicated that every $Mg^{2+}$-binding site in the CP region (Mg2–Mg7) contributes to the formation of the MgtE closed state (*Hattori et al., 2009*).

Although the structure of $Mg^{2+}$-free full-length MgtE has not been reported, we have presented the crystal structure of the N-terminal 275 residues of MgtE, which includes the whole CP region (residues 1–263) in both the $Mg^{2+}$-free and -bound states (*Hattori et al., 2007b*). A structural comparison of the two states suggests that $Mg^{2+}$ binding alters the relative orientation of the N and CBS domains in each subunit, as well as changing the inter-helical angle between the plug helices of the two subunits. Recently, a nuclear magnetic resonance (NMR) analysis of the MgtE CP region in the $Mg^{2+}$-free state, conducted by our laboratory, revealed that the N domain tumbles widely in space, and that it transiently approaches the CBS domain (*Imai et al., 2012*). Based on these results, we have proposed a structural mechanism for MgtE gating, in which $Mg^{2+}$ binding to the CP region of MgtE allosterically alters the conformation of the plug helices, resulting in gate closure. However, the concentration of $Mg^{2+}$ required to saturate each $Mg^{2+}$-binding site remains unknown, as does the site(s) to which $Mg^{2+}$-binding causes the changes in the structure and dynamics of each portion of MgtE, resulting in the closed MgtE state.

In the present study, we used solution NMR spectroscopy to investigate $Mg^{2+}$-dependent changes in the structure and dynamics of full-length MgtE. $Mg^{2+}$-titration experiments using the wild-type and mutant forms of MgtE revealed the functional roles of each $Mg^{2+}$-binding site: Whereas the Mg1, Mg2, Mg3, and Mg6 sites exhibit a higher affinity for $Mg^{2+}$ and stabilize the structure of the TM region, the Mg4, Mg5, and Mg7 sites play critical roles in changing the conformation and dynamics of the CP region including the plug helices, leading to channel closure. This study thus revealed the $Mg^{2+}$-dependent gating mechanism of MgtE that underlies cellular $Mg^{2+}$ homeostasis.

## Results

### Methyl-TROSY signals of Ile δ1 methyl groups as probes of the conformation and dynamics of MgtE

Full-length MgtE was overexpressed in *Escherichia coli*, solubilized in *n*-dodecyl-β-maltoside (DDM) micelles, and purified to homogeneity (*Figure 2—figure supplement 1*). Size exclusion chromatography analysis indicated an apparent molecular weight of approximately 160 kDa, suggesting that MgtE formed a dimer in the micelles (*Figure 2—figure supplement 2*).

In order to observe the solution NMR signals with sufficiently high resolution and sensitivity in such a large membrane protein, we elected to observe the methyl-TROSY spectra of Ile δ1 methyl groups in {u-$^2$H, Ileδ1-[$^{13}$CH$_3$]}MgtE. This demonstrated that the fifteen Ile residues in full-length MgtE are widely distributed across the whole MgtE structure (*Figure 1B*), with I28 and I84 being located in the N-terminal domain; I168, I171, I190, I201, and I242 in the CBS domain; I260 in the plug helix; and I291, I293, I302, I338, I339, I397, and I438 in the TM region. Hereafter, each Ile residue is represented by its residue number and its location, such as I28$^N$, I168$^{CBS}$, I260$^{Plug}$, and I291$^{TM}$.

We next established the resonance assignments for twelve of the fifteen Ile δ1 methyl groups in the $Mg^{2+}$-free and bound states, respectively, by comparing the methyl-TROSY spectra of fifteen individual Ile to Val mutants with that of the wild-type protein, in the absence and presence of 16 mM $Mg^{2+}$ (*Figure 2* and *Table 1*). The remaining three signals for I242$^{CBS}$, I291$^{TM}$, and I302$^{TM}$ were assigned only in the presence of 16 mM $Mg^{2+}$, due to line broadening and/or degeneracy of the signals in the absence of $Mg^{2+}$. It should be also noted that $Mg^{2+}$-induced NMR spectral changes reflect the changes in the conformation and dynamics at each Ile site, but not the direct binding of $Mg^{2+}$ to the Ile residues. This is because the distances between the Ile Cδ1 atoms and $Mg^{2+}$ are longer than 8.0 Å in the crystal structure of the $Mg^{2+}$-bound MgtE (PDB code: 2ZY9), which is sufficiently longer than the sum of the radii of a methyl group (1.6 Å) and a hydrated $Mg^{2+}$ (4.8 Å).

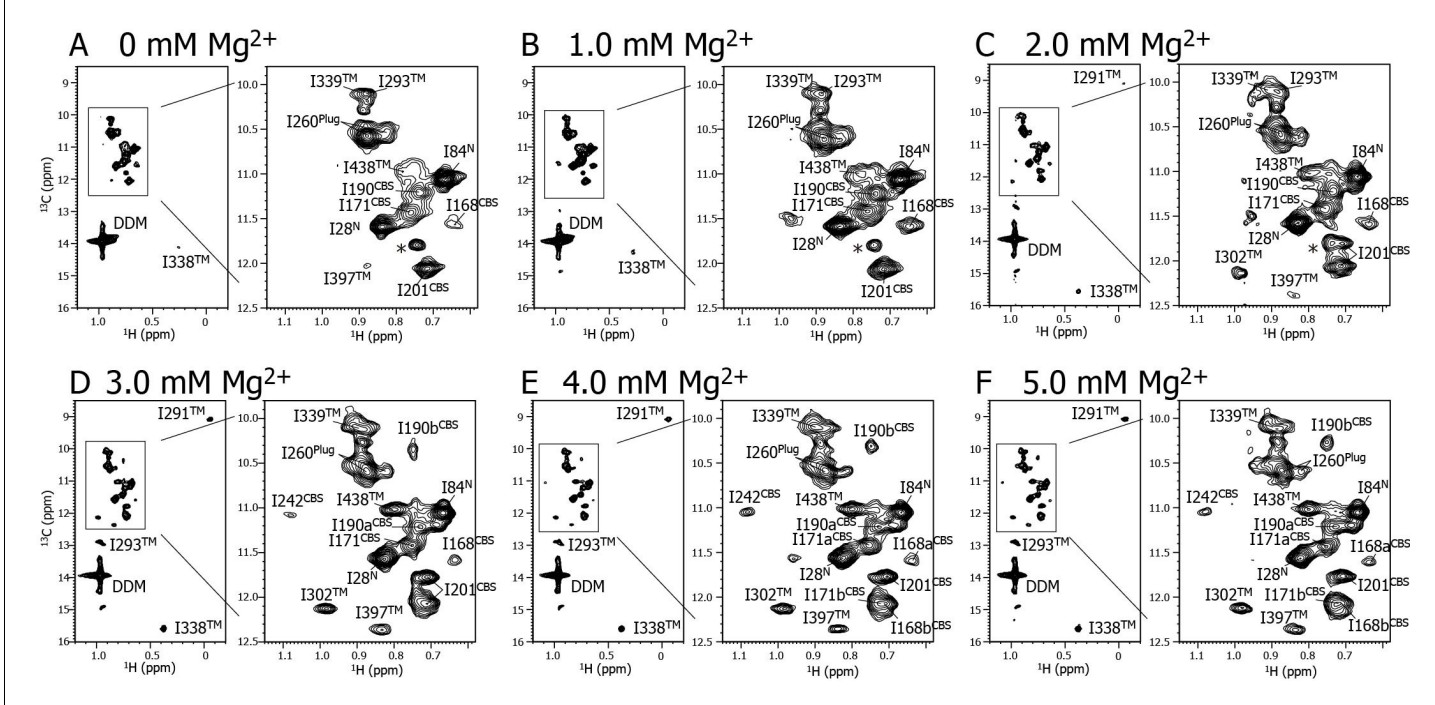

**Figure 2.** Mg²⁺ titration experiments monitored by NMR signals of the Ile δ1 methyl groups. Methyl-TROSY spectra of {u-²H, Ileδ1-[¹³CH₃]} MgtE reconstituted into DDM micelles in the presence of (**A**) 0, (**B**) 1.0, (**C**) 2.0, (**D**) 3.0, (**E**) 4.0, and (**F**) 5.0 mM Mg²⁺. The signal with an asterisk is unassigned.

DOI: https://doi.org/10.7554/eLife.31596.004

The following figure supplements are available for figure 2:

**Figure supplement 1.** Purification of MgtE.
DOI: https://doi.org/10.7554/eLife.31596.005

**Figure supplement 2.** Analysis of the oligomeric state of purified full-length MgtE in DDM micelles.
DOI: https://doi.org/10.7554/eLife.31596.006

## Mg²⁺-induced changes in the conformation and dynamics of MgtE as probed by Ile NMR signals

*Figure 2* shows a series of methyl-TROSY spectra at Mg²⁺ concentrations ranging from 0 to 5.0 mM. At 0 mM Mg²⁺ (*Figure 2A*), three markedly strong signals (I28$^N$, I84$^N$, and I260$^{Plug}$) were observed, together with nine weak and/or broad signals (I168$^{CBS}$, I171$^{CBS}$, I190$^{CBS}$, I201$^{CBS}$, I293$^{TM}$, I338$^{TM}$, I339$^{TM}$, I397$^{TM}$, and I438$^{TM}$). However, signals for I242$^{CBS}$, I291$^{TM}$, and I302$^{TM}$ were not observed.

As the Mg²⁺ concentration increased (*Figure 2B–2F*), different types of changes were observed, which are summarized in *Figure 3*. The three strong signals (I28$^N$, I84$^N$, and I260$^{Plug}$; boxed in red in *Figure 3*) exhibited decreased intensities, whereas three new signals appeared for the three residues previously unobserved at 0 mM Mg²⁺ (I242$^{CBS}$, I291$^{TM}$, and I302$^{TM}$; boxed in yellow in *Figure 3*). In addition, among the nine weak and/or broad signals, six signals exhibited chemical shift changes: I397$^{TM}$ and I438$^{TM}$ in a fast-to-intermediate exchange regime, and I201$^{CBS}$, I293$^{TM}$, and I338$^{TM}$ in a slow exchange regime (*Figure 2*, and cyan in *Figure 3*). The other three (I168$^{CBS}$, I171$^{CBS}$, I190$^{CBS}$) of the nine weak and/or broad signals became split in the Mg²⁺-bound state (*Figure 2*, and magenta in *Figure 3*). Only the signal for I339$^{TM}$ exhibited no significant change.

The NMR spectral changes apparently saturated at 4.0 mM Mg²⁺, above which the NMR spectra were essentially identical (see below for details). It has been reported that at Mg²⁺ concentration of 4 mM, or above, the MgtE channel closes (*Hattori et al., 2009*), suggesting that the NMR spectra at Mg²⁺ concentrations lower, or higher, than 4 mM reflect the open or closed states of MgtE, respectively.

**Table 1.** Assignments of Ile δ1 methyl groups of full-length MgtE.
Chemical shifts for δ1 methyl groups in Ile residues in the $Mg^{2+}$-free and bound states are shown. Labels a and b are used to discriminate two signals from a single Ile residue. $^1H$ chemical shifts were referenced to external sodium 2,2-dimethyl-2-silapentane-5-sulfonate (0 ppm), and $^{13}C$ chemical shifts were referenced indirectly. Spectral width (eight ppm) in the $^{13}C$ dimension was subtracted from the $^{13}C$ chemical shift of I338 owing to spectral aliasing.

| Methyl group | $Mg^{2+}$-free state (0 mM $Mg^{2+}$) | | $Mg^{2+}$-bound state (5.0 mM $Mg^{2+}$) | |
| --- | --- | --- | --- | --- |
| | $^1H$ (ppm) | $^{13}C$ (ppm) | $^1H$ (ppm) | $^{13}C$ (ppm) |
| I28 | 0.729 | 14.16 | 0.714 | 14.13 |
| I84 | 0.559 | 13.63 | 0.526 | 13.62 |
| I168a | 0.539 | 14.12 | 0.526 | 14.14 |
| I168b | – | – | 0.606 | 14.62 |
| I171a | 0.651 | 13.99 | 0.644 | 13.98 |
| I171b | – | – | 0.606 | 14.62 |
| I190a | 0.627 | 13.77 | 0.621 | 13.77 |
| I190b | – | – | 0.641 | 12.83 |
| I201 | 0.606 | 14.62 | 0.599 | 14.33 |
| I242 | – | – | 0.85 | 14.13 |
| I260 | 0.771 | 13.13 | 0.762 | 13.15 |
| I291 | – | – | −0.171 | 11.63 |
| I293 | 0.773 | 12.66 | 0.858 | 15.45 |
| I302 | – | – | 0.872 | 14.68 |
| I338 | 0.145 | 8.67 | 0.264 | 10.16 |
| I339 | 0.799 | 12.66 | 0.798 | 12.65 |
| I397 | 0.771 | 14.59 | 0.728 | 14.92 |
| I438 | 0.674 | 13.54 | 0.693 | 13.58 |

DOI: https://doi.org/10.7554/eLife.31596.007

## $Mg^{2+}$-concentration dependence of the NMR spectral changes

In order to investigate $Mg^{2+}$-concentration dependence, the changes in the signal intensities and the chemical shifts for the well-resolved signals were plotted against $Mg^{2+}$ concentration (*Figure 4A and B*). The plots showed sigmoid curves that were well-fitted with a Hill's equation, resulting in the $Mg^{2+}$ concentration reaching the half maximal values of the changes ($[Mg^{2+}]_{1/2}$) of 0.8–2.5 mM, with Hill coefficient (*n*) values of 1.8–7.6. These results indicate that $Mg^{2+}$ binding to multiple sites cooperatively affects the conformation and dynamics of the Ile residues at different locations in the MgtE molecule.

*Figure 4C* shows an overlay of the plots that normalize the change from the $Mg^{2+}$-free state to the $Mg^{2+}$-bound state; the Ile residues in the TM and other regions are shown in the upper and lower panels, respectively. Clearly, three Ile residues in the TM region (I302$^{TM}$, I397$^{TM}$, I438$^{TM}$) and I260$^{Plug}$ experienced $Mg^{2+}$-induced changes at $Mg^{2+}$ concentrations of 0–2 mM, whereas the other Ile residues were affected at 2–3 mM $Mg^{2+}$. The functionally more important $Mg^{2+}$ concentrations are those at which the NMR spectral changes saturate (hereafter, referred to as $[Mg^{2+}]_{sat}$), above which each Ile residue adopts the closed MgtE conformation. The normalized plots in *Figure 4C* indicate that the apparent $[Mg^{2+}]_{sat}$ values are 3 mM for the Ile residues in the plug helix and the TM region, and 4 mM for those in the N and CBS domains.

## Role of each $Mg^{2+}$ binding site in the $Mg^{2+}$-induced conformational change in MgtE

Our previous electrophysiological investigation indicated that $Mg^{2+}$-binding sites, Mg2, Mg3, Mg5 and Mg7 in the CP region contribute to the formation of the closed state, in addition to Mg1 in the

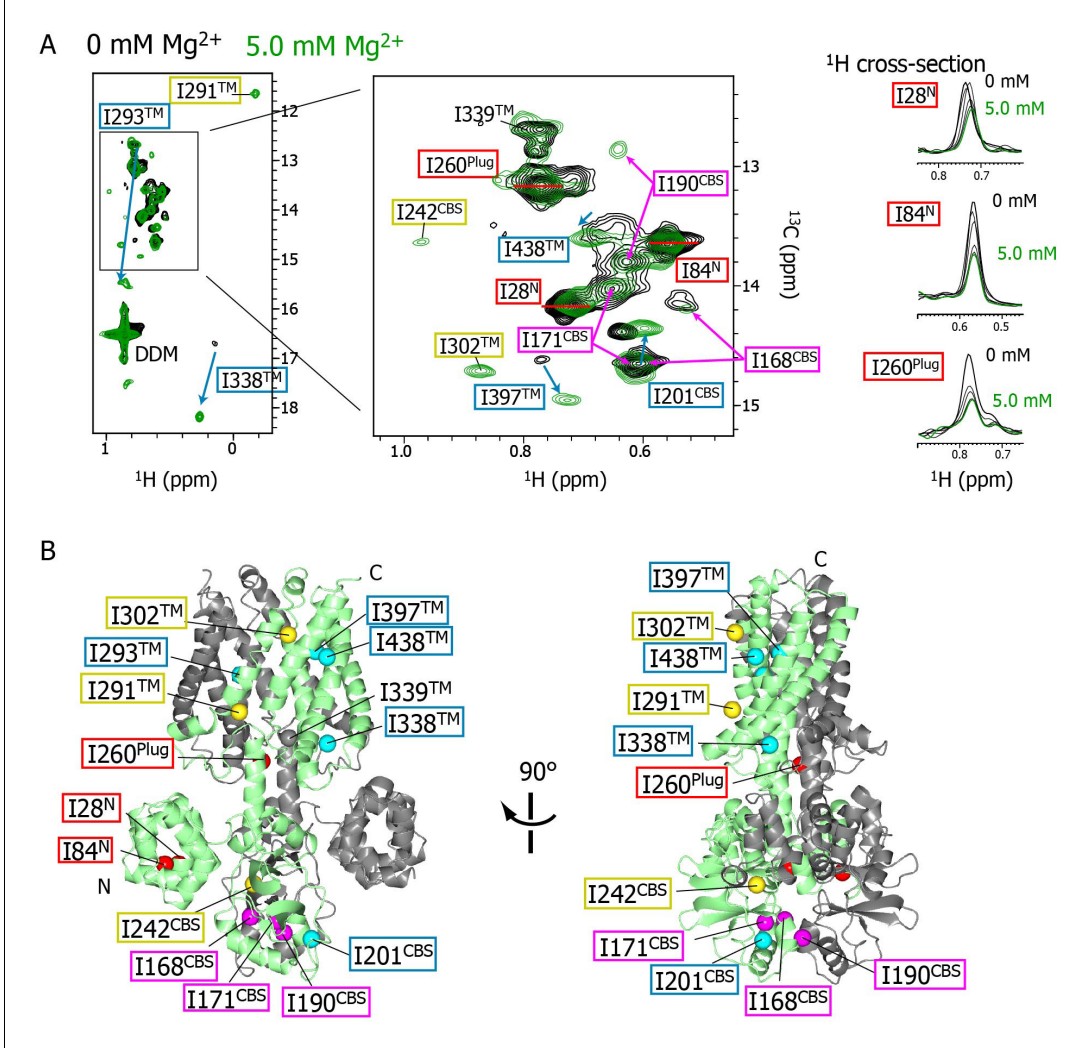

**Figure 3.** $Mg^{2+}$-concentration-dependent spectral changes in MgtE. (A) Superposition of the methyl-TROSY spectra in the $Mg^{2+}$-free state (0 mM; black) and the bound state (5.0 mM; green). Upon $Mg^{2+}$ binding, (i) the residues whose signals broadened are boxed in red, (ii) the residues whose signals appeared or were sharpened are boxed in yellow, (iii) the residues exhibiting chemical shift changes are boxed in cyan, with an arrow indicating the direction of the chemical shift changes, and (iv) the residues for which two signals were observed (split signals) are boxed in magenta. The right panels show the $^1H$ cross-sections for the I28N, I84N, and I260Plug signals at $Mg^{2+}$ concentrations of 0, 1.0, 2.0, 3.0, 4.0, and 5.0 mM. (B) Mapping of the residues whose signals exhibited the $Mg^{2+}$-dependent changes in (A) on the $Mg^{2+}$-bound crystal structure (PDB code:2ZY9). Ile $C\delta1$ atoms are coloured as in (A), whereas the Ile $C\delta1$ atom of I339, which exhibited no significant change, is coloured grey.

DOI: https://doi.org/10.7554/eLife.31596.008

TM region (*Hattori et al., 2009*). We then examined the role of the rest of $Mg^{2+}$-binding sites, Mg4 and Mg6, in the MgtE channel activity. Although wild-type closed at 10 mM $Mg^{2+}$ on the periplasmic side, D91A/D247A mutant for Mg4 and D95A mutant for Mg6 did not close even at 20 mM $Mg^{2+}$, which is similar to D226N/D250A mutant for Mg5 analysed previously (*Figure 5—figure supplement 1*, panels A and B). *Figure 5—figure supplement 1C* summarizes the $Mg^{2+}$-concentration dependence of the open probability of the wild-type and the $Mg^{2+}$-binding site mutants of MgtE, which was shown in the current and previous study (*Hattori et al., 2009*). These results indicate that mutation of any one of the $Mg^{2+}$-binding sites, Mg2-Mg7, prevents the formation of the conformation in the closed state. In other words, every $Mg^{2+}$-binding site, Mg2-Mg7, is required for the $Mg^{2+}$-dependent closure of MgtE channel.

In order to uncover the role of $Mg^{2+}$ binding to each site in the structural changes related to channel closure, we prepared seven mutants in which, based on the MgtE crystal structure, each of

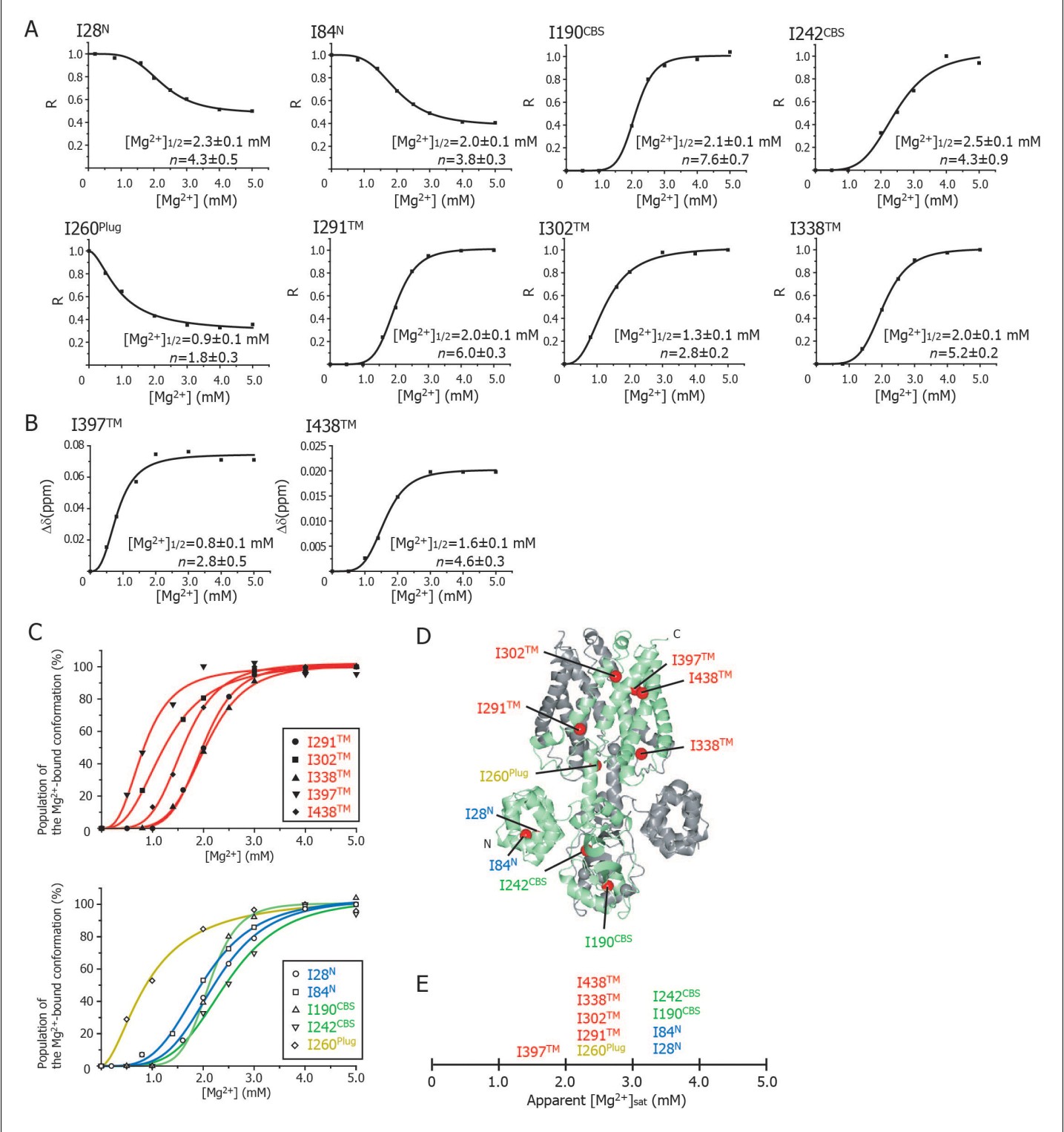

**Figure 4.** Cooperative effects of Mg$^{2+}$ binding on Ileδ1 methyl signals. (**A**) Normalized signal intensities (**R**) and (**B**) chemical shift changes (Δδ) of Ileδ1 methyl signals plotted against Mg$^{2+}$ concentration ([Mg$^{2+}$]). The solid lines represent the best fit curves of a Hill equation to the data. The estimated Mg$^{2+}$ concentrations reaching the half maximal values of the changes ([Mg$^{2+}$]$_{1/2}$) and the Hill coefficients (**n**) are shown in each graph. (**C**) Superposition of the changes in the population of the Mg$^{2+}$-bound conformation of the Ile δ1 methyl signals. The normalized changes in the chemical shift values and signal intensities in A and B were regarded as changes in the population of the Mg$^{2+}$-bound conformation. The solid lines, coloured blue for the N domain, green for the CBS domain, yellow for the plug helix, and red for the TM region, represent the best fit curves of Hill equations to the data. The

*Figure 4 continued on next page*

Figure 4 continued

graphs of Ile residues in the TM and other regions are separated for clarity. (D) The positions of Ile residues (red) subjected to the analyses. (E) Apparent saturated $Mg^{2+}$ concentrations, $[Mg^{2+}]_{sat}$ at which the population of the $Mg^{2+}$-bound conformation is above 90%. Apparent $[Mg^{2+}]_{sat}$ is 3 mM for the Ile residues in the plug helix and the TM region, and 4 mM for the Ile residues in the N and CBS domains in the CP region.

DOI: https://doi.org/10.7554/eLife.31596.009

the sites (Mg1 to Mg7) was disabled by mutation of its $Mg^{2+}$-coordinated acidic residue(s), and investigated the effect of the mutations on the $Mg^{2+}$-dependent NMR spectral changes (**Figure 5**). The locations of Mg1–Mg7 are indicated in the structure in **Figure 1B**, and are schematically depicted in **Figure 6A**. Since the spectra of these mutants in the $Mg^{2+}$-free state are essentially identical to that of the wild-type, these mutations introduced no structural change in MgtE in the $Mg^{2+}$-free state (**Figure 5—figure supplement 2**).

**Figure 5A** shows the methyl-TROSY spectra of the D432A mutant, in which the $Mg^{2+}$-binding of Mg1 is impaired, in the presence of 4.0 mM $Mg^{2+}$ (red), superimposed on the spectra of the wild-type MgtE in the presence (**Figure 5A**, left, black) or absence (**Figure 5A**, middle, black) of 4.0 mM $Mg^{2+}$. The NMR spectra of the mutant (hereafter, referred to as the Mg1-binding mutant) differs from that of the wild-type in the presence of 4.0 mM $Mg^{2+}$, as the signals for I190$^{CBS}$, I201$^{CBS}$, I242$^{CBS}$, I291$^{TM}$, I302$^{TM}$, I338$^{TM}$, I397$^{TM}$, and I438$^{TM}$ did not appear in the former (**Figure 5A**). These differences in the signals from the CBS domain and the TM region suggest that, in the presence of a saturating amount of $Mg^{2+}$, the conformation of these regions in the Mg1-binding mutant is different from that in the wild-type.

In addition, the markedly strong signal intensities seen for I28$^N$, I84$^N$, and I260$^{Plug}$ in the Mg1-binding mutant did not decrease as the $Mg^{2+}$ concentration increased (**Figure 5A**, right), whereas those in wild-type MgtE significantly decreased in a $Mg^{2+}$-dependent manner as shown in **Figure 3A**. These results indicate that, in the presence of 4.0 mM $Mg^{2+}$, the conformation and dynamics of the N domain and the plug helix in the Mg1-binding mutant also differ from those of the wild-type MgtE.

In contrast, the spectrum of the Mg1-binding mutant in the presence of 4 mM $Mg^{2+}$ is similar to that of the wild-type at 0 mM $Mg^{2+}$ (**Figure 5A**, middle). The only differences are the missing of the signals for I168$^{CBS}$, I338$^{TM}$, I397$^{TM}$, and I438$^{TM}$, which were observed for the wild-type at 0 mM $Mg^{2+}$. The missing of these signals is presumably caused by the exchange broadening between $Mg^{2+}$-bound and unbound states for the sites Mg2-7 of the Mg1-binding mutant. These spectral comparisons strongly suggest that the mutation of the Mg1-binding site precludes the $Mg^{2+}$-induced conformational changes in the whole MgtE molecule, as has been observed for the wild-type MgtE (**Figure 6B**, left).

Essentially the same spectra, seen for the Mg1-binding mutant (D432A), were observed for the Mg2- (E258Q), Mg3- (D259N), and Mg6- (D95N) binding mutants (**Figure 5—figure supplement 3**), suggesting that the mutation at the site Mg2, Mg3, or Mg6 also precludes the complete $Mg^{2+}$-induced conformational changes in the whole MgtE molecule at 4 mM $Mg^{2+}$. Therefore, the binding mutants for Mg2, Mg3, and Mg6 would all be expected to require higher $Mg^{2+}$ concentration to saturate $Mg^{2+}$-binding to all the sites that are not mutated.

These results indicate that $Mg^{2+}$ binding to the Mg1, Mg2, Mg3, and Mg6 sites is required for the $Mg^{2+}$-induced changes in the conformation and dynamics of the whole MgtE molecule (**Figure 6B**, left). Conversely, the Mg4- (D91A/D247A), Mg5- (D226N/D250A), and Mg7- (E59A) binding mutants exhibited $Mg^{2+}$-induced spectral changes in the Ile residues in the TM region (I291$^{TM}$, I302$^{TM}$, I338$^{TM}$, I397$^{TM}$, and I438$^{TM}$) identical to those seen in the wild-type protein, whereas the spectral changes for the Ile residues in the N domain (I28$^N$ and I84$^N$), the CBS domain (I190$^{CBS}$ and I242$^{CBS}$) and the plug helix (I260$^{Plug}$) differed from the wild-type spectra (**Figure 5B–D**, respectively).

For the Mg4-binding mutant (D91A/D247A), the $Mg^{2+}$-induced change in I190$^{CBS}$, which was seen in the wild-type protein, was not observed (**Figure 5B**). The $Mg^{2+}$ concentrations, at which the intensity changes of the I28$^N$ and I84$^N$ signals saturated, which reflects the population of molecules in the $Mg^{2+}$-bound conformation, were significantly higher (>10 and >25 mM, respectively, estimated from curve fitting of the Hill's equation) compared to 4 mM for the wild-type (**Figure 5E**). The

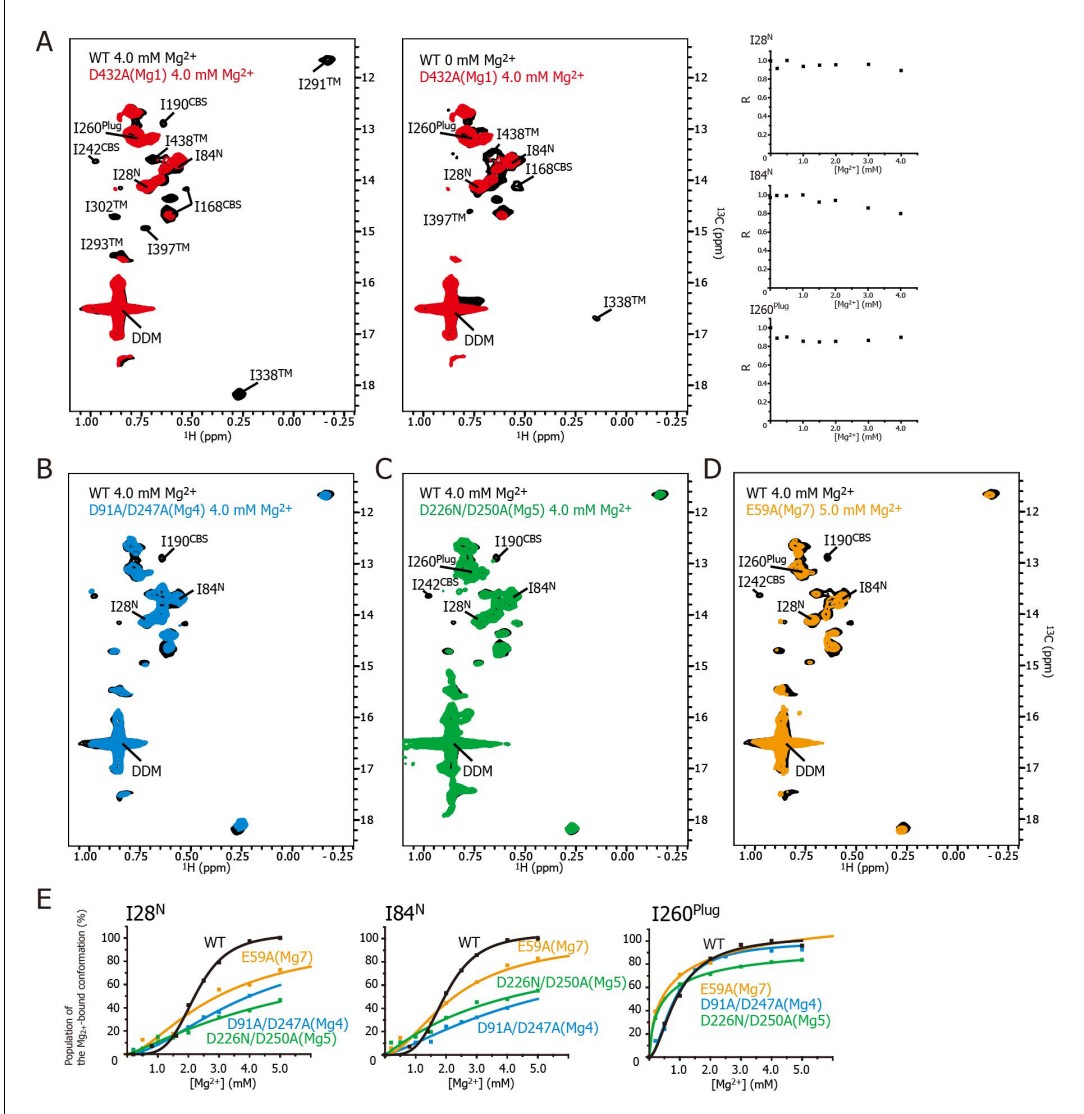

**Figure 5.** Effects of mutations at each $Mg^{2+}$-binding site on the $Mg^{2+}$-dependent conformational changes. (**A**) Methyl-TROSY spectrum of the D432A mutant for Mg1 in the presence of 4.0 mM $Mg^{2+}$ (red) superimposed on the spectrum of the wild-type (WT) protein (black) in the presence (left) and absence (middle) of 4.0 mM $Mg^{2+}$. Normalized signal intensities (**R**) for $I28^N$, $I84^N$, and $I260^{Plug}$ in the Mg1 mutant were plotted against $Mg^{2+}$ concentration ($[Mg^{2+}]$). (**B–D**) Methyl-TROSY spectra of the D91A/D247A mutant of Mg4 (B, cyan), the D226N/D250A mutant of Mg5 (C, green), and the E59A mutant of Mg7 (D, orange) in the presence of 4.0–5.0 mM $Mg^{2+}$, are overlaid on that of the wild-type (black) in the presence of 4.0 mM $Mg^{2+}$. The signals differing from those in the wild-type are labelled. (**E**) The population of the $Mg^{2+}$-bound conformation of $I28^N$, $I84^N$, and $I260^{Plug}$ plotted against $Mg^{2+}$ concentrations ($[Mg^{2+}]$). The solid lines are coloured black for the wild-type, cyan for the Mg4 mutant (D91A/D247A), green for Mg5 mutant (D226N/D250A), and orange for Mg7 mutant (E59A), respectively.

DOI: https://doi.org/10.7554/eLife.31596.010

The following figure supplements are available for figure 5:

**Figure supplement 1.** Patch-clamp analyses of wild-type and mutant forms of MgtE using the $Mg^{2+}$-auxotrophic *E. coli* strain.

DOI: https://doi.org/10.7554/eLife.31596.011

**Figure supplement 2.** Effects of the mutations at each $Mg^{2+}$-binding site on the MgtE structure in the absence of $Mg^{2+}$.

DOI: https://doi.org/10.7554/eLife.31596.012

**Figure supplement 3.** Effects of the mutations at each $Mg^{2+}$ binding site on the $Mg^{2+}$-dependent changes in MgtE structure and dynamics.

DOI: https://doi.org/10.7554/eLife.31596.013

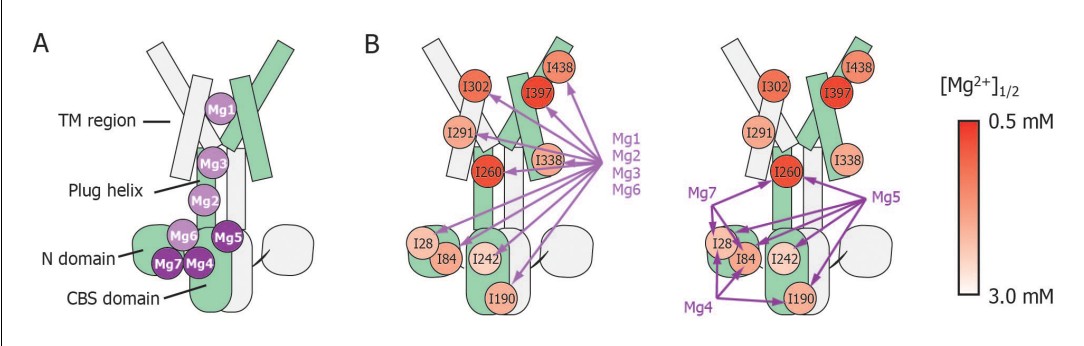

**Figure 6.** Schematic summary of the effects of mutations at each Mg$^{2+}$-binding site on Mg$^{2+}$-dependent changes in the conformation and dynamics. (A) Positions of Mg$^{2+}$-binding sites in one subunit. (B) Relationship between the Mg$^{2+}$-binding sites and the Ile residues whose Mg$^{2+}$-dependent changes of the NMR signal were affected by the mutations at each Mg$^{2+}$-binding site. The mutated Mg$^{2+}$-binding sites and the affected residues are connected with arrows. The Ile residues are coloured by the [Mg$^{2+}$]$_{1/2}$ values shown in **Figure 4**, as indicated.
DOI: https://doi.org/10.7554/eLife.31596.014

changes in the other Ile residues were identical to those in the wild-type protein. These results suggest that Mg$^{2+}$ binding to the Mg4-binding site affects the structure and dynamics of the N and CBS domains, as shown by I28$^N$, I84$^N$, and I190$^{CBS}$ (**Figure 6B**, right).

For the Mg5-binding mutant (D226N/D250A), the Mg$^{2+}$-induced changes in I190$^{CBS}$ and I242$^{CBS}$ were not observed (**Figure 5C**). The Mg$^{2+}$ concentrations, at which the intensity changes of the I28$^N$, I84$^N$ and I260$^{Plug}$ signals saturated, were significantly higher (>30, >30, and >8 mM, respectively), than the wild-type (4 mM) (**Figure 5E**). In addition, this mutant exhibited lower cooperativity for I260$^{Plug}$ ($n$ = 0.8) than that of the wild type ($n$ = 1.8). These results suggest that Mg$^{2+}$ binding to the Mg5 site elicits changes in the structure and dynamics of the N and CBS domains, as well as the plug helix, as shown by I28$^N$, I84$^N$, I190$^{CBS}$, I242$^{CBS}$, and I260$^{Plug}$ (**Figure 6B**, right).

For the Mg7-binding site mutant (E59A), the majority of the Ile signals, except for I28$^N$, I84$^N$ and I260$^{Plug}$, exhibited Mg$^{2+}$-dependent changes that were identical to those in the wild-type (**Figure 5D**). The Mg$^{2+}$ concentrations, at which the intensity changes of the I28$^N$ and I84$^N$ signals saturated, were significantly higher (>10 and>7 mM, respectively), than the wild-type (4 mM) (**Figure 5E**). This mutant also exhibited lower cooperativity for I260$^{Plug}$ ($n$ = 0.6) than that of the wild type ($n$ = 1.8). These results suggest that Mg$^{2+}$ binding to the Mg7 binding site elicits changes in the N domain and the plug helix, as shown by I28$^N$, I84$^N$, and I260$^{Plug}$ (**Figure 6B**, right).

Overall, whereas Mg$^{2+}$ binding to the Mg4-, Mg5-, and Mg7-binding sites does not affect Ile residues in the TM region, all affect the N domain Ile residues (**Figure 6**). Channel closure requires Mg$^{2+}$ binding to all of the sites, Mg1–Mg7 (**Hattori et al., 2009**), suggesting that Mg$^{2+}$ binding to the Mg4, Mg5, and Mg7 sites causes channel closure via Mg$^{2+}$-dependent structural changes in the N domain.

## Discussion

In this study, we characterized the effects of Mg$^{2+}$ binding on the structure and dynamics of MgtE in a site-specific manner, by observing the NMR signals of Ile δ1 methyl groups that are distal to specific Mg$^{2+}$-binding sites. Notably, the spectral changes do not reflect Mg$^{2+}$-binding directly but rather indicate changes in the conformation and dynamics of Ile residues from which the Mg$^{2+}$-binding effects could be inferred. Solution NMR method is known as one of the methods to solve the three-dimensional structure of relatively small proteins. For larger proteins, NMR spectral changes caused by the functionally related stimuli such as ligand binding reflect the changes in the conformation and dynamics that are related to their functions. Together with the high resolution crystal structure of Mg$^{2+}$-bound, closed state of MgtE (**Hattori et al., 2009**), Mg$^{2+}$-dependent changes of the methyl-TROSY spectra could provide the information which sites of MgtE change its structure and dynamics between the Mg$^{2+}$-saturated, closed state and the Mg$^{2+}$-free or partially bound, open state.

## Mg$^{2+}$-dependent changes in the structure and dynamics of MgtE

The purified MgtE protein showed Mg$^{2+}$-dependent spectral changes that were mostly saturated at 4.0 mM Mg$^{2+}$. Based on a previous electrophysiological result that MgtE closes at cytoplasmic Mg$^{2+}$ concentrations above 5–10 mM (*Hattori et al., 2009*), the NMR spectrum of MgtE above 4 mM is likely to reflect the conformation of MgtE in its closed state. Although the crystal structure of MgtE in the Mg$^{2+}$-free state is not available, current NMR spectra provides structural information regarding the Mg$^{2+}$-free MgtE and its changes upon Mg$^{2+}$-binding.

In the Mg$^{2+}$-free state, the NMR signals for I242$^{CBS}$, I291$^{TM}$, and I302$^{TM}$ were very broad, whereas I28$^{N}$, I84$^{N}$, and I260$^{Plug}$ had sharp, strong signals (*Figure 2A*). Together with our previous paramagnetic relaxation enhancement study, indicating that the N domain undergoes free tumbling motions in the Mg$^{2+}$-free state (*Imai et al., 2012*), these findings suggest that Mg$^{2+}$-free MgtE has a conformational equilibrium with different kinetics between the TM/CBS region and N/Plug region. The conformational equilibrium is suppressed by Mg$^{2+}$-binding, as evidenced by sharpening/appearance of the weak and/or broad signals, and by broadening of the very sharp signals for I28$^{N}$, I84$^{N}$, and I260$^{Plug}$, resulting in the formation of the closed state of MgtE.

## The effects of Mg$^{2+}$ binding at each site on the conformation and dynamics of MgtE

Mg$^{2+}$-titration experiments for the wild-type and mutant forms of MgtE indicated that all the Ile residues in the TM region (Ile$^{TM}$) were affected by Mg$^{2+}$ binding to the Mg1, Mg2, Mg3, and Mg6 sites, whereas mutation of the Mg4-, Mg5-, or Mg7-binding sites did not affect the Mg$^{2+}$-induced changes in the structure and dynamics of the TM region (*Figures 5* and *6*, and *Figure 5—figure supplement 3*). In addition to saturation of the changes for Ile$^{TM}$ at 3 mM Mg$^{2+}$ (*Figure 4*), Mg$^{2+}$ binding to the Mg1, Mg2, Mg3 and Mg6 sites is suggested to saturate at 3 mM, which completes the Mg$^{2+}$-dependent conformational changes in the TM region.

Based on the fact that mutation at the Mg1-, Mg2-, Mg3-, or Mg6-binding sites also affected Mg$^{2+}$-dependent changes in Ile residues in the CP region, Mg$^{2+}$ binding to these sites is required for the structural/dynamic changes in the CP region as well as in the TM region. It should be noted that the structural changes in the TM region seem to occur in two steps (*Figure 4*). The residues I302$^{TM}$, I397$^{TM}$, and I438$^{TM}$, which lie in the proximity of Mg1 at the extracellular side of the TM region, change at lower Mg$^{2+}$ concentrations. On the other hand, the residues I291$^{TM}$ and I338$^{TM}$, which locate at the intracellular side of the TM region, change at the Mg$^{2+}$ concentrations similar to the changes of the Ile residues in the N and CBS domains. The structural changes in two steps seem to reflect that the latter residues at the intracellular side of the TM region are also affected by the Mg$^{2+}$-binding to the Mg$^{2+}$ sites, Mg2, Mg3 and/or Mg6 in the intracellular domains.

Conversely, Mg$^{2+}$ binding to the remainder of the sites, Mg4, Mg5, and Mg7, regulates Ile residues only in the CP region consisting of the N and CBS domains as well as the plug helix, with no effect on the TM region. Specifically, binding at Mg4 affects Ile residues in the N and CBS domains, binding at Mg5 affects Ile residues in the N and CBS domains, as well as the plug helix, and binding at Mg7 affects Ile residues in the N domain and the plug helix (*Figures 5* and *6*, and *Figure 5—figure supplement 3*). As shown in *Figure 4*, Mg$^{2+}$-dependent changes in the Ile residues in the CP region saturate at 4 mM, strongly suggesting that Mg$^{2+}$ binding to the Mg4, Mg5, and Mg7 sites saturates at 4 mM. This changes the conformation of the CP region including the N and CBS domains, resulting in the formation of the Mg$^{2+}$-bound closed state, as observed in the crystal structure (*Hattori et al., 2007b*, *Hattori et al., 2009*).

## Contribution of Mg4, Mg5, and Mg7 to gate closure

In the crystal structure of the full-length MgtE in the Mg$^{2+}$-bound form (*Hattori et al., 2007b*, *Hattori et al., 2009*), Mg$^{2+}$ bound at the Mg4 site forms a bridge between the N domain and the plug helix in the same subunit, whereas Mg$^{2+}$ bound at the Mg5 and Mg7 sites forms inter-subunit bridges between the CBS domain and the plug helix, and between the N and CBS domains, respectively (schematically shown in *Figure 7*, right). Mg$^{2+}$ binding to these sites neutralizes the electrostatic repulsion of the acidic residues forming the Mg$^{2+}$-binding sites in the N and CBS domains and the plug helix, enabling the cooperative formation of a compact globular conformation in the closed state of MgtE.

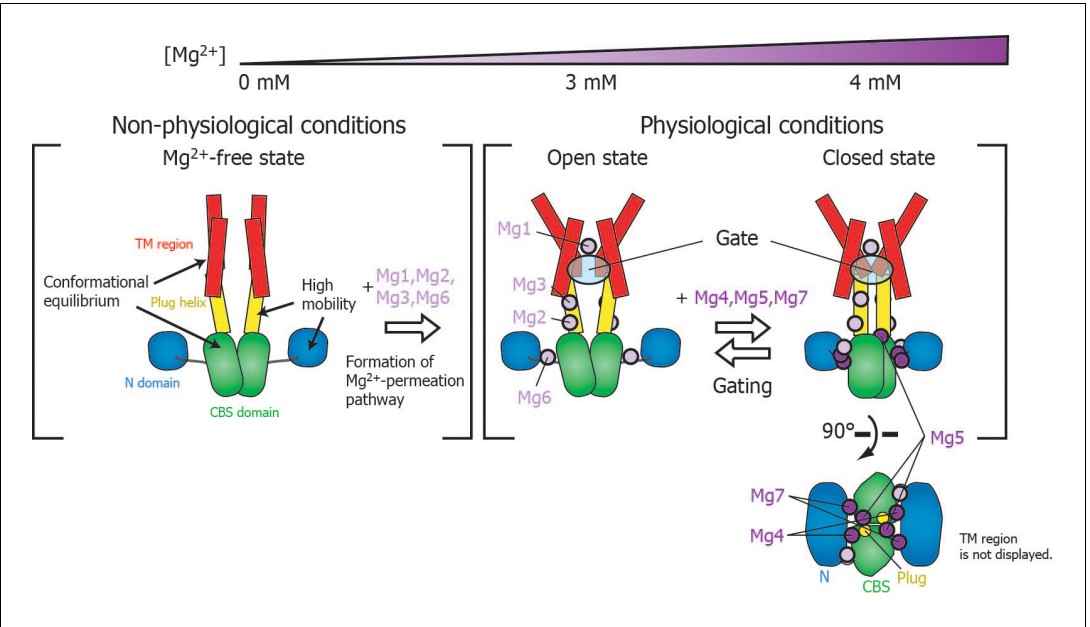

**Figure 7.** Roles for each Mg$^{2+}$ binding site in the gating of MgtE. In the Mg$^{2+}$-free state, which MgtE does not adopt in physiological conditions, conformational equilibrium exists between the CBS domain and the TM region, whereas the N domain and the plug helix have high motional flexibility. Cooperative Mg$^{2+}$ binding to Mg1, Mg2, Mg3, and Mg6 form the ion-conducting pore, whose gate is closed by cooperative Mg$^{2+}$ binding to Mg4, Mg5, and Mg7. The Mg$^{2+}$ bound at Mg4 forms a bridge between the N domain and the plug helix in the same subunit. The Mg$^{2+}$ bound at Mg5 forms a bridge between the CBS domain in one subunit and the plug helix in the other subunit. The Mg$^{2+}$ bound at Mg7 forms a bridge between the N domain in one subunit and the CBS domain in the other subunit. Together, these bound Mg$^{2+}$ atoms may stabilize the domain orientations in close proximity to each other.

DOI: https://doi.org/10.7554/eLife.31596.015

The closed gate is formed and stabilized by the interaction between N424$^{TM}$ and L263$^{Plug}$ on the cytoplasmic side of the ion-conducting pore in the TM region (*Figure 1—figure supplement 1*) (*Hattori et al., 2009*). Thus, the I260$^{Plug}$ NMR signal, which is located in the C-terminal region of the plug helix, might serve as a good probe for the conformational change of the gate region. Although the fast motion of I260$^{Plug}$ is mostly suppressed at 3 mM (*Figure 4*), Mg$^{2+}$ binding to Mg5 and Mg7, which affects I260$^{Plug}$, saturates at 4 mM (*Figure 6B*). Therefore, the complete conformational change of I260$^{Plug}$ is likely to be achieved at 4 mM, which contributes to the stabilization of the closed gate.

## Physiological relevance of MgtE gating

*Figure 7* schematically summarizes a model of the Mg$^{2+}$-dependent gating of MgtE. The present study revealed that gating of MgtE occurs at Mg$^{2+}$ concentrations of 4 mM. Below this threshold, MgtE adopts an open state, and allows the transport of extracellular Mg$^{2+}$ into cells; thus, the intracellular Mg$^{2+}$ level is maintained higher than the threshold. Consequently, under physiological conditions, MgtE is not in the Mg$^{2+}$-free state. Cooperative Mg$^{2+}$ binding to the Mg1-, Mg2-, Mg3-, and Mg6-binding sites, which changes the conformation and dynamics of the TM region, saturates at 3 mM, suggesting that these sites are also constitutively Mg$^{2+}$-bound at the physiological Mg$^{2+}$ level, which is higher than 4 mM.

Electrophysiological results indicated that a mutation at Mg1 abolishes Mg$^{2+}$ conductivity, whereas a mutation at either of the sites Mg2, Mg3, or Mg6 does not (*Hattori et al., 2009* and *Figure 5—figure supplement 1C*). These results are apparently inconsistent with the current NMR data that the Mg$^{2+}$-binding to the Mg1-, Mg2-, Mg3-, and Mg6-binding sites is cooperative, i.e., the loss of the Mg$^{2+}$-binding ability at either of the Mg2-, Mg3-, or Mg6-binding site, should impair the Mg$^{2+}$-binding affinity of the Mg1-binding site, and therefore, should abolish the ion conductivity.

This inconsistency is presumably due to the differences in the experimental conditions on the $Mg^{2+}$ concentration between NMR and electrophysiological experiments. Whereas the $Mg^{2+}$ concentration is constant in an NMR sample solution, ranging from 0 to 5 mM, those in the electrophysiological experiments are 90 mM and 0.2 mM, respectively, on the extracellular and intracellular sides. Among the seven $Mg^{2+}$-binding sites, the Mg1-binding site is the only one among the seven sites that lies at the extracellular side of the TM region of MgtE (*Figure 1B*). This suggests that, under the condition for the electrophysiological experiments, high $Mg^{2+}$ concentration at 90 mM on the extracellular side seems to force the Mg1-binding site of the Mg2-, Mg3-, and Mg6-binding mutants to accommodate a $Mg^{2+}$ ion, leading to the channel opening of these mutants, in spite of the cooperativity revealed by NMR.

The mutation at either of the Mg2-, Mg3-, or Mg6-binding sites prevents the channel closure in the presence of 10–20 mM $Mg^{2+}$ (*Figure 5—figure supplement 1*). Therefore, the roles of the cooperative $Mg^{2+}$ binding to the Mg1-, Mg2-, Mg3-, and Mg6-binding sites are likely to include the formation of an ion-conducting pore in the TM region mainly by the Mg1-binding site, and adjustment of the $Mg^{2+}$-binding affinities of the Mg4, Mg5, and Mg7 sites, enabling the CP region to act as an $Mg^{2+}$ sensor for gating in response to increases in intracellular $Mg^{2+}$ levels above 4 mM. The functional roles of each of these $Mg^{2+}$-binding sites therefore underlie the intracellular $Mg^{2+}$ homeostasis mediated by MgtE.

## Conclusion

NMR analysis in combination with a high resolution crystal structure has provided site-specific information on changes in the protein conformation and dynamics of MgtE in relation to its function. By translating this information to SLC41 family members, human orthologues of the prokaryotic MgtE protein (*Goytain and Quamme, 2005a*; *Kolisek et al., 2008*; *Moomaw and Maguire, 2008*; *Sahni and Scharenberg, 2013*), it may be possible to identify a binding site for a novel ligand, which could be used to develop a novel treatment for diseases caused by abnormal $Mg^{2+}$ levels, including cardiovascular disease, diabetes, and high blood pressure (*Alexander et al., 2008*).

# Materials and methods

**Key resources table**

| Reagent type (species) or resource | Designation | Source or reference | Identifiers | Additional information |
|---|---|---|---|---|
| Gene (*Thermus thermophilus*) | MgtE | doi: 10.1107/S1744309107032332 | Uniprot ID: Q5SMG8 | |
| Strain, strain background (*Escherichia coli*) | BW25113 ΔmgtA ΔcorA ΔyhiD DE3 | doi: 10.1038/emboj.2009.288 | | |
| Recombinant DNA reagent | pET28a-MgtE | doi: 10.1107/S1744309107032332 | | |

## Plasmid construction and expression

We utilised a plasmid encoding full-length MgtE from *T. thermophilus* with an N-terminal His × 6 tag and an HRV-3C protease recognition site (*Hattori et al., 2007a*). The MgtE mutant constructs were generated through polymerase chain reaction-based mutagenesis. All MgtE proteins were expressed in *E. coli* C41 (DE3) cells. {u-$^2$H, Ileδ1-[$^{13}CH_3$]}MgtE was expressed according to a previous study (*Tugarinov et al., 2006*).

## Sample preparation

MgtE and its mutants were purified as follows. The harvested cells were suspended in a buffer containing 50 mM HEPES-NaOH (pH 7.0), 150 mM NaCl and 20 mM imidazole, supplemented with 1 mM phenylmethylsulfonyl fluoride and lysed by sonication followed by centrifugation at 1,750 × *g* for 10 min. The supernatants were then ultra-centrifuged at 100,000 × *g* for 30 min. The pellet was solubilized for 2 hr at 277 K with a buffer containing 50 mM HEPES-NaOH (pH 7.0), 150 mM NaCl, 40 mM n-dodecyl -D-maltoside) (DDM), and 20 mM imidazole. After centrifugation at 14,000 × *g* for 30 min, the supernatant was applied to a TALON column (Clontech, Mountain View, CA, USA). After washing the column with a buffer containing 50 mM HEPES-NaOH (pH 7.0), 150 mM NaCl, 1 mM

DDM, and 20 mM imidazole, the protein was eluted with the same buffer supplemented with 150 mM imidazole. The N-terminal His × 6 tag was then cleaved using the HRV-3C protease. The cleaved His × 6 tag, undigested MgtE, and HRV-3C protease were removed by passing the sample through a HIS-select column (Sigma, St. Louis, MO, USA).

For the NMR experiments, the sample buffer was exchanged with NMR buffer (20 mM HEPES-NaOH (pH 7.2), 20 mM NaCl, 100% $D_2O$). The pH value of the 100% $D_2O$ buffer was calibrated by adding 0.4 pH unit to the reading on the pH meter (*Blanchard, 1984*).

## NMR spectroscopy

NMR spectra were observed at 313 K on Bruker Avance 500, 600, or 800 MHz spectrometers equipped with a cryogenic probe. For $Mg^{2+}$-titration experiments, small aliquots of the NMR buffer containing 20–200 mM $MgCl_2$ were added. The MgtE concentration was 200 µM, whereas the DDM was approximately 10 mM, estimated from the signal intensity of the DDM methyl signal in $^1H$ 1D spectra.

The assignments of the Ile δ1 methyl-TROSY signals in the $Mg^{2+}$-free and bound states were obtained by site-directed mutagenesis. We observed methyl-TROSY spectra for fifteen individual Ile mutants, in which each Ile residues was mutated to Val (i.e. I28V, I84V, I168V, I171V, I190V, I201V, I242V, I260V, I291V, I293V, I302V, I338V, I339V, I397V, or I438V). The spectra of these mutants were compared to those of the wild-type protein in the $Mg^{2+}$-free and $Mg^{2+}$-bound states, respectively. The missing signals in the spectra of the Ile mutants were assigned as the signals arising from the mutated Ile residues.

Chemical shift differences, Δδ, were calculated using the equation:

$$\Delta\delta = \left\{ (\Delta\delta_{1H}) + (\Delta\delta_{13C}/5.8)^2 \right\}^{0.5}$$

where $\Delta\delta_{1H}$ and $\Delta\delta_{13C}$ are the chemical shift differences in the $^1H$ and $^{13}C$ dimensions, respectively. The normalised factor (5.8) was determined from the ratio of the variance of methyl $^1H$ and $^{13}C$ chemical shifts, deposited in the Biological Magnetic Resonance Data Bank. The signal intensities of Ileδ1 methyl signals for a series of the titration spectra were normalised based on the intensity of the DDM methyl signal. The $Mg^{2+}$ concentration reaching the half maximal values in the changes ($[Mg^{2+}]_{1/2}$) and Hill coefficients ($n$) were calculated using the equations

$$\Delta = \Delta_{max}[Mg^{2+}]^n / \left( [Mg^{2+}]_{1/2}^n + [Mg^{2+}]^n \right)$$

or

$$\Delta' = 1 - \Delta'_{max}[Mg^{2+}]^n / \left( [Mg^{2+}]_{1/2}^n + [Mg^{2+}]^n \right)$$

where Δ and Δ' are the changes in the chemical shift or the signal intensity of Ile δ1 methyl signals.

## Patch-clamp analysis

The $Mg^{2+}$-auxotrophic *E. coli* strain (BW25113 ΔmgtA ΔcorA ΔyhiD DE3) was transformed with each MgtE expressing plasmid, and maintained in growth medium supplemented with 100 mM $MgSO_4$. *E. coli* giant spheroplasts were prepared as described previously (*Hattori et al., 2009*). Spheroplasts expressing wild-type and mutant forms of MgtE were plated on glass coverslips in a bath solution containing 200 mM *N*-methyl-D-glucamine, 90 mM $MgCl_2$, 300 mM glucose and 10 mM HEPES (pH 7.2). Borosilicate pipettes (Harvard Apparatus, Kent, UK), with a resistance of 5–8 MΩ, were filled with a pipette solution (250 mM *N*-methyl-D-glucamine, 90 mM $MgCl_2$, 300 mM glucose and 10 mM HEPES (pH 7.2). After gigal seal formation, a patch of membrane was excised and the bath solution was exchanged with a batch solution containing 290 mM *N*-methyl-D-glucamine, 0.2 mM $MgCl_2$, 300 mM glucose and 10 mM HEPES (pH 7.2). The membrane patch voltage was clamped and currents were recorded using an Axopatch 200B amplifier (Axon CNS, Molecular Devices), coupled to an A/D converter (Axon CNS, Molecular Devices) and controlled by the pclamp10 software (Axon CNS, Molecular Devices). Currents were filtered at 2 kHz and sampled at 5 kHz.

## Acknowledgements

This work was supported in part by grants from the Japan New Energy and Industrial Technology Development Organization (NEDO) and the Ministry of Economy, Trade, and Industry (METI), and the Japan Agency for Medical Research and Development (AMED) (Grant name: Development of core technologies for innovative drug development based upon IT, to IS.), a Grant-in-Aid for Scientific Research on Priority Areas from the Japanese Ministry of Education, Culture, Sports, Science, and Technology (to IS), Japan Society for the Promotion of Science KAKENHI Grant Numbers JP16H01368 and JP17H03978 (to MO), a grant from The Vehicle Racing Commemorative Foundation (to MO), and a grant from SENSHIN Medical Research Foundation (to MO). We would like to thank Editage (www.editage.jp) for English language editing.

## Additional information

### Funding

| Funder | Grant reference number | Author |
|---|---|---|
| New Energy and Industrial Technology Development Organization | | Ichio Shimada |
| Ministry of Economy, Trade and Industry | | Ichio Shimada |
| Japan Agency for Medical Research and Development | | Ichio Shimada |
| Ministry of Education, Culture, Sports, Science, and Technology | | Ichio Shimada |
| Japan Society for the Promotion of Science | KAKENHI JP16H01368 | Masanori Osawa |
| Vehicle Racing Commemorative Foundation | | Masanori Osawa |
| SENSHIN Medical Research Foundation | | Masanori Osawa |
| Japan Society for the Promotion of Science | KAKENHI JP17H03978 | Masanori Osawa |

The funders had no role in study design, data collection and interpretation, or the decision to submit the work for publication.

### Author contributions

Tatsuro Maruyama, Conceptualization, Investigation, Visualization, Writing—original draft, Writing—review and editing; Shunsuke Imai, Conceptualization, Supervision, Investigation; Tsukasa Kusaki-zako, Koichi Ito, Resources, Writing—review and editing; Motoyuki Hattori, Ryuichiro Ishitani, Conceptualization, Resources; Osamu Nureki, Conceptualization, Resources, Project administration; Andrès D Maturana, Conceptualization, Investigation, Visualization, Writing—review and editing; Ichio Shimada, Masanori Osawa, Conceptualization, Supervision, Funding acquisition, Visualization, Writing—original draft, Project administration, Writing—review and editing

### Author ORCIDs

Motoyuki Hattori https://orcid.org/0000-0002-5327-5337
Ryuichiro Ishitani http://orcid.org/0000-0002-4136-5685
Osamu Nureki http://orcid.org/0000-0003-1813-7008
Masanori Osawa http://orcid.org/0000-0002-1285-4316

### Decision letter and Author response

Decision letter https://doi.org/10.7554/eLife.31596.020
Author response https://doi.org/10.7554/eLife.31596.021

## Additional files

### Supplementary files

• Transparent reporting form
DOI: https://doi.org/10.7554/eLife.31596.016

### Major datasets

The following dataset was generated:

| Author(s) | Year | Dataset title | Dataset URL | Database, license, and accessibility information |
|---|---|---|---|---|
| Maruyama T, Imai S, Kusakizako T, Hattori M, Ishitani R, Nureki O, Ito K, Maturana AD, Shimada I, Osawa M | 2018 | Functional roles of Mg2+ binding sites in ion-dependent gating of a Mg2+ channel, MgtE, revealed by solution NMR | https://datadryad.org//resource/doi:10.5061/dryad.hd575 | Available at Dryad Digital Repository under a CC0 Public Domain Dedication |

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
