## [Decision Letter]

Thank you for submitting your article "Functional roles of Mg^2+^ binding sites in ion-dependent gating of a Mg^2+^ channel, MgtE, revealed by solution NMR" for consideration by *eLife*. Your article has been favorably evaluated by Richard Aldrich (Senior Editor) and three reviewers, one of whom is a member of our Board of Reviewing Editors. The following individual involved in review of your submission has agreed to reveal their identity: Charles R Sanders (Reviewer #2).

The reviewers have discussed the reviews with one another and the Reviewing Editor has drafted this decision.

This is an interesting paper on the role of the binding of seven Mg(II) ions to the microbial MgtE channel, providing the apparent Mg(II) binding affinity as well as Hill coefficients informing on binding cooperativity. This was a technically challenging project in which a combination of sophisticated isotopic labeling methods and methyl-TROSY methods allowed NMR spectra to be acquired for some 15 assigned isoleucine residues (spread rather evenly throughout the protein). This took quite a lot of work for a >150 kDa detergent/intact channel complex. Binding was monitored examining the Mg(II)-concentration dependence of peak positions and/or intensity. Different peaks exhibited changes that led to different apparent Kd and Hill coefficients. Additional NMR studies conducted using Mg(II) ion binding site mutants, led to the realization that three of the Mg(II) sites form a distinct set of sites from the other 4 sites. In particular, the three sites are proposed to be responsible for stabilizing the open form of the channel, whereas the other 4 may help stabilize the channel but also exert other modulating effects, perhaps leading to channel closure at higher ion concentrations.

The reviewers agreed that the work is interesting and important. However, they felt that relating Mg(II) binding results to function would be critical to establish the role of the bound ions. In particular, it was pointed out that the binding data and structural inferences that suggest highly cooperative ion binding should be reconciled with existing functional data (e.g., Figure 2 of Nature Comm. 8: 148 (2017)), which show no cooperativity in the channel response to increased Mg(II) concentration. Furthermore, similar functional data on mutants targeting Mg(II) binding sites should help resolve the role of the groups of the sites. For example, mutations of sites 4, 5, and 7 should lead to a channel that does not close at elevated Mg(II) concentrations.

*Reviewer #1:*

The manuscript by Osawa et al. describes an elegant study of a bacterial magnesium channel MgtE (homologous to eukaryotic SLC41), which is responsible for the maintenance of the intracellular Mg levels. The channel is tightly regulated by Mg, and closes at Mg concentrations above 5 mM. Previous crystal structures showed that the dimeric channel contains one central Mg ion in the permeation pathway (Mg1) and 6 additional sites per subunit in three cytoplasmic regions: N-domain, CBS domain and plug helix. To address the role of the Mg binding sites, the authors employed Ile Methyl-TROSY NMR. They found that in the absence of Mg, N-domain and plug residue I260 showed high mobility. Titrating in Mg, led to changes of Ile peaks characteristics throughout the protein with EC50 varying between 0.8 and 2.5 mM, and with high Hill coefficients from 2 to 7. The authors mutated out each Mg binding site one at a time, and found Mg binding to sites Mg1, 2, 3 and 6 was essential for all conformational changes. In contrast, binding to sites Mg4, 5, and 7 was not essential for changes in TM regions, but was required for changes in N-domain, CBS domain, and plug helix (Mg 5 and 7). The authors construct a model whereby at low, but physiologically relevant Mg concentrations, Mg binds cooperatively to sites 1, 2, 3 and 6 to produce a conducting state; at higher concentrations Mg binds cooperatively to sites 4, 5 and 7, increasing interactions between N-domain and CBS and leading to channel closing. I found the work compelling even if there were some unresolved questions.

1) While the spectra of WT apo protein and Mg1 mutant in the presence of 4 mM Mg are similar as the authors state, they are not identical. The authors should discuss what are the differences and what might be their origin. In Discussion (subsection “The effects of Mg^2+^ binding at each site on the conformation and dynamics of MgtE”, first paragraph), the authors say that Mg binding to sites 1, 2, 3 and 6 affect Ile in TM region "cooperatively". The last term is ambiguous here. In particular, the TM Ile sites show Mg-dependent changes with different EC50s and Hill coefficients, such that, for example, I397 has EC50 of 0.8 (Nhill=2.8) while I338 has EC50 of 2 mM (Nhill=5.2). The latter residue shows sensitivity more similar to residues in the N- and CBS-domains. Therefore, while Mg binding to the sites might be cooperative, the structural changes in TM region may occur in two steps (with binding of Mg to the first and second sets of binding sites). This needs to be clarified.

2) According to Figure 5E, Mg4 mutations do not affect Mg-dependent changes of I260Plug, while both Mg7 and Mg5 mutations decrease cooperativity, but not affinity. Text in the sixth and seventh paragraphs of the subsection “Role of each Mg^2+^ binding site in the Mg^2+^-induced conformational change in MgtE” contradicts the figure (also Figure 6B, right).

3) The statement about cooperative binding to sites 4, 5, 7 is confusing. Binding to site 4 does not seem to affect the structural changes in the plug, while binding to sites 5 and 7 does.

4) Can the signal changes, such as the chemical shift changes and peaks' sharpening/appearance be interpreted in terms of changes of protein dynamics?

5) What kind of cooperative coupling is expected between the subunits in the dimer?

*Reviewer #2:*

This is an interesting paper on the role of the binding of seven Mg(II) ions to the microbial MgtE channel, providing the apparent Mg(II) binding affinity as well as Hill coefficients informing on binding cooperativity. This was a technically challenging project in which a combination of sophisticated isotopic labeling methods and methyl-TROSY methods allowed NMR spectra to be acquired for some 15 assigned isoleucine residues (spread rather evenly throughout the protein). This took quite a lot of work for a >150 kDa detergent/intact channel complex. Binding was monitored examining the Mg(II)-concentration dependence of peak positions and/or intensity. Different peaks exhibited changes that led to different apparent Kd and Hill coefficients. Along with the results of additional NMR studies conducted using Mg(II) ion binding site mutant, this led to the realization that three of the Mg(II) sites can regarded a distinct set of site from the other 4 sites. In particular, the three sites are proposed to be responsible for stabilizing the gated form of the channel, whereas the other 4 may help stabilize the channel but also exert other structural and/or dynamic roles in modulating Mg(II) channel function. Exactly what these other roles is was a little vague. This is an admirable and difficult study that significantly advances our understanding of Mg(II) channel structure and function. I especially admire the care with which they quantitative the Mg(II)-dependence of the very modest changes in NMR peaks positions/intensities.

This well written paper could be published essentially as is. However, the degree to which this NMR study established exactly what these two sets of Mg(II) binding events are doing in terms of channel function would be greatly enhanced by accompanying [Mg(II)]-dependent channel function measurements for both WT and the mutants they characterized by NMR. Such measurements are not reported in this paper. I would strongly encourage the authors to consider revising this paper to include the results of functional measurements (perhaps provided by expert collaborators). Without this, I view this paper as borderline in terms of providing enough new insight into MgtE channel function to justify publication in a journal with the stature of *eLife*.

*Reviewer #3:*

Magnesium is a ubiquitous ion necessary for many biological functions with several unique chemical properties that differentiate it from other divalent cations such as Calcium. These include a very high energy of dehydration, small ionic but very large hydrated radius. How ion channels and transporters move Mg^2+^ in and out of compartments is important to understanding both the molecular mechanisms and function of this ion. The authors of this study use solution-NMR to study the dynamics of the prokaryotic Mg^2+^ channel, MgtE. The size and helix content make this a challenging NMR target. Addressing this, the group used methyl-TROSY with specific labeling of isoleucines as reporters of channel dynamics in detergent micelles. Coupled with site directed mutagenesis of residues responsible for binding Mg^2+^ residues (determined from previous functional and structural studies), the authors initially titrate and then use a set Mg^2+^ concentration to study what role binding plays in the dynamics of MgtE.

The authors have used NMR in previous studies to begin to address MgtE dynamics using PRE. Further, they've characterized the function of MgtE. Working from these previous studies, the authors combine this information to derive a model that asks how the dynamics of the transmembrane region, N-domain, CBS, and net cytoplasmic portion are related to the known Mg^2+^ binding sites. They arrive at a model where 4 of 7 Mg^2+^ sites act on the transmembrane domains and Plug Helix, whereas the remaining 3 of 7 Mg^2+^ sites are principally focused on dynamics in the cytoplasmic domain. They propose a model based on the absence or presence of Mg^2+^ on channel function. This is a useful finding that advances our knowledge of how the structure, dynamics and function reveal the mechanisms of Mg^2+^ permeation in an ion channel.

Questions/concerns:

1) Physiological range – While 0.05% of dry weight mass of *E. coli* is Mg^2+^, and the estimated cellular concentrations are 30 mM, the free Mg^2+^ is estimated to be around 0.3 mM for *E. coli*. How does the model in Figure 7 account for this range and for the estimated binding affinities?

2) It is suggested that the authors establish an explicit correlation between the behavior of the NMR-derived Mg^2+^ binding behavior at the different Mg^2+^-binding sites of MgtE, and the know Mg^2+^ dependence of the channel open probability, as determined by electrophysiological methods (perhaps as part of Figure 5?).

[Editors' note: further revisions were requested prior to acceptance, as described below.]

Thank you for resubmitting your work entitled "Functional roles of Mg^2+^ binding sites in ion-dependent gating of a Mg^2+^ channel, MgtE, revealed by solution NMR" for further consideration at *eLife*. Your revised article has been favorably evaluated by Richard Aldrich (Senior Editor) and a Reviewing Editor.

The manuscript has been improved but there are some remaining issues that need to be addressed before acceptance, as outlined below:

The main issue that the reviewers raised during the initial review was the lack of clear correspondence between the conformational responses of the protein to binding of Mg ions as detected by NMR and the functional behavior. The authors have addressed the criticism by providing functional data on mutations targeting every Mg binding site (either previously published or conducted specifically for this study). The data are succinctly summarized in the figure in their point-by-point response. What is still missing is putting these data in perspective of the current NMR study. In other words, how do these data agree (or disagree) with Figure 7? This is clearly a very complicated system, and perhaps not all data are expected to be in perfect correspondence, but key aspects should be addressed. For example, what is the role of sites 2, 3 and 5 if their mutations are not detrimental for channel opening but are for channel closing? I would have liked the authors to include the figure into the paper and discuss clearly to what extent their proposed model is or is not fully consistent with functional data.

---

## [Author Response]

The reviewers agreed that the work is interesting and important. However, they felt that relating Mg(II) binding results to function would be critical to establish the role of the bound ions. In particular, it was pointed out that the binding data and structural inferences that suggest highly cooperative ion binding should be reconciled with existing functional data (e.g., Figure 2 of Nature Comm. 8: 148 (2017)), which show no cooperativity in the channel response to increased Mg(II) concentration.

The patch clamp analysis in our recent paper (Tomita et al. Nature Comm. 2017) indicated that open probability decreases in response to the elevation of the Mg^2+^ level. This is consistent with our NMR results that detect the Mg^2+^-dependent changes in the population of Mg^2+^-bound conformation of I260^Plug^ (Figure 4C), which serves as a good probe for the conformational change of the gate region (N424^TM^ and L263^Plug^).

The conformation of I260^Plug^ is affected by the Mg^2+^-binding to Mg1, 2, 3, 5, 6 and 7 (Figure 6). The Mg^2+^-binding to these sites differentially affects the conformation of I260^Plug^, resulting in the small Hill coefficient value of 1.8 ± 0.3. This weak cooperativity might not be observed in the patch clamp analysis, due to the differences in the experimental conditions, such as high salt concentration, existence of the Mg^2+^-concentration gradient and membrane potential across the lipid bilayer for the patch clamp analysis, whereas low ion strength of 20 mM NaCl is suitable to detect the Mg^2+^-binding effects by NMR with high sensitivity.

Furthermore, similar functional data on mutants targeting Mg(II) binding sites should help resolve the role of the groups of the sites. For example, mutations of sites 4, 5, and 7 should lead to a channel that does not close at elevated Mg(II) concentrations.

Thank you very much for the suggestion. The functional data for the sites, 1, 2, 3, 5, and 7 were previously reported (Hattori et al., 2009), as follows:

Mg1-binding mutant, D432A, exhibited no channel activity.

Mg2-binding mutant, E258Q, did not close at 20 mM Mg^2+^.

Mg3-binding mutant, D259N, did not close at 20 mM Mg^2+^.

Mg5-binding mutant, D226N/D250A, did not close at 20 mM Mg^2+^.

Mg7-binding mutant, E59A, did not close at 10 mM Mg^2+^, but closed at 20 mM Mg^2+^.

Here, we have performed the functional experiments for the binding mutants for the sites 4 and 6 (Figure 5—figure supplement 1 in the revised manuscript), indicating that these mutants do not close at 20 mM Mg^2+^. We described this as follows, “We then examined the role of the rest of Mg^2+^-binding sites, Mg4 and Mg6, in the MgtE channel activity. Although wild-type closed at 10 mM Mg^2+^ on the periplasmic side, D91A/D247A mutant for Mg4 and D95A mutant for Mg6 did not close even at 20 mM Mg^2+^, which is similar to D226N/D250A mutant for Mg5 analysed previously (Figure 5—figure supplement 1). […] In other words, every Mg^2+^-binding site, Mg2-Mg7, is required for the Mg^2+^-dependent closure of MgtE channel.”

Reviewer #1:

[…] 1) While the spectra of WT apo protein and Mg1 mutant in the presence of 4 mM Mg are similar as the authors state, they are not identical. The authors should discuss what are the differences and what might be their origin.

As the reviewer pointed out, there are differences in these two spectra: the signals for I168^CBS^, I338^TM^, I397^TM^, and I438^TM^ were observed for the WT apo protein (Figure 5A, middle, black), whereas they were not observed for the Mg1 mutant in the presence of 4 mM Mg^2+^(Figure 5 A, middle, red). As the signals for these residues were observed at different chemical shifts for the WT protein in the presence of 4 mM Mg^2+^ (Figure 5A, left, black), missing of these signals in the spectrum for the Mg1-binding mutant in the presence of 4 mM Mg^2+^ suggests that these signals were broadened due to the chemical exchange among multiple conformations, caused by the partial and inhomogeneous binding of Mg^2+^ to the sites Mg2-Mg7.

The description regarding the differences has been added as follows, “The only differences are the missing of the signals for I168^CBS^, I338^TM^, I397^TM^, and I438^TM^, which were observed for the wild-type at 0 mM Mg^2+^. The missing of these signals is presumably caused by the exchange broadening between Mg^2+^-bound and unbound states for the sites Mg2-7 of the Mg1-binding mutant.”

In Discussion (subsection “The effects of Mg^2+^ binding at each site on the conformation and dynamics of MgtE”, first paragraph), the authors say that Mg binding to sites 1, 2, 3 and 6 affect Ile in TM region "cooperatively". The last term is ambiguous here. In particular, the TM Ile sites show Mg-dependent changes with different EC50s and Hill coefficients, such that, for example, I397 has EC50 of 0.8 (Nhill=2.8) while I338 has EC50 of 2 mM (Nhill=5.2). The latter residue shows sensitivity more similar to residues in the N- and CBS-domains. Therefore, while Mg binding to the sites might be cooperative, the structural changes in TM region may occur in two steps (with binding of Mg to the first and second sets of binding sites). This needs to be clarified.

Thank you for the comment. The mutations at either site of Mg1, 2, 3, or 6 caused the similar changes in the NMR signals for the identical set of the Ile residues (Figure 5A and Figure 5—figure supplement 3), suggesting that the Mg^2+^-binding to either of these sites contributes to the structural changes of these Ile residues. However, as the reviewer pointed out, the structural changes in the TM region of the wild type protein seem to occur in two steps (Figure 4).

Whereas the residues I302^TM^, I397^TM^, and I438^TM^, which lie in the proximity of Mg1 at the extracellular side of the TM region, change at lower Mg^2+^ concentrations, the residues I291^TM^ and I338^TM^, which locate at the intracellular side of the TM region, change at the Mg^2+^ concentrations similar to the changes of the Ile residues in the N and CBS domains. The structural changes in two steps seem to reflect that the latter residues at the intracellular side of the TM region are also affected by the Mg^2+^-binding to the intracellular domains.

Accordingly, we revised the Discussion, as follows,

“cooperatively” has been removed.

We have removed “appears to be cooperative and”.

We have added a paragraph, “It should be noted that the structural changes in the TM region seem to occur in two steps (Figure 4). […] The structural changes in two steps seem to reflect that the latter residues at the intracellular side of the TM region are also affected by the Mg^2+^-binding to the Mg^2+^ sites, Mg2, Mg3 and/or Mg6 in the intracellular domains.”

2) According to Figure 5E, Mg4 mutations do not affect Mg-dependent changes of I260Plug, while both Mg7 and Mg5 mutations decrease cooperativity, but not affinity. Text in the sixth and seventh paragraphs of the subsection “Role of each Mg^2+^ binding site in the Mg^2+^-induced conformational change in MgtE” contradicts the figure (also Figure 6B, right).

We are grateful for the reviewer’s comment. As the reviewer pointed out, the Mg5- and Mg7-binding mutants decreased the cooperativity of I260^Plug^. We revised the ninth paragraph of the subsection “Role of each Mg^2+^ binding site in the Mg^2+^-induced conformational change in MgtE” to state that Mg5 and Mg7 decreased the cooperativity of I260^Plug^, and line 274 to state that Mg7 also affects the plug helix. In addition, we revised the right panel of Figure 6B.

3) The statement about cooperative binding to sites 4, 5, 7 is confusing. Binding to site 4 does not seem to affect the structural changes in the plug, while binding to sites 5 and 7 does.

We agree with the reviewer’s comment and have revised the text as follows, “Although the fast motion of I260^Plug^ is mostly suppressed at 3 mM (Figure 4), Mg^2+^ binding to Mg5 and Mg7, which affects I260^Plug^, saturates at 4 mM (Figure 6B).”

4) Can the signal changes, such as the chemical shift changes and peaks' sharpening/appearance be interpreted in terms of changes of protein dynamics?

Yes, they can be. Both signal line width and the way of the chemical shift changes reflect protein dynamics. Basically, a nuclear spin with a fast tumbling motion gives a sharp NMR signal, whereas a spin tumbling slowly gives a broad signal. The rate of the chemical exchange between conformations A and B as compared to difference in the resonance frequencies for the conformations A and B also affects the appearance of the signals (Cavanagh J. et al., “Protein NMR Spectroscopy (Second Edition) Principles and Practice ISBN: 978-0-12-164491-8).

5) What kind of cooperative coupling is expected between the subunits in the dimer?

The Mg^2+^ at the site 1 bridges the TM regions of the two subunits, forming the Mg^2+^-permeation pathway at the interface of the two subunits in the TM region. As described in the last paragraph of the subsection “The effects of Mg^2+^ binding at each site on the conformation and dynamics of MgtE”, the N domains tumble fast in the apo state, and they interact with the two CBS domains in the Mg^2+^-bound state. Mg^2+^ at the site 5 and 7 bridges the CBS domain in one subunit with the plug helix and the N domain of another subunit, respectively, which are assisted by Mg^2+^ at the site 4 and 6 that bridge the N and CBS domains in the same subunit. Thus, cooperative coupling is expected between the subunits in the dimer, which is achieved by the inter-subunit bridging by the Mg^2+^ at the sites 1, 5, and 7 directly (Figure 7).

Reviewer #2:[…] This well written paper could be published essentially as is. However, the degree to which this NMR study established exactly what these two sets of Mg(II) binding events are doing in terms of channel function would be greatly enhanced by accompanying [Mg(II)]-dependent channel function measurements for both WT and the mutants they characterized by NMR. Such measurements are not reported in this paper. I would strongly encourage the authors to consider revising this paper to include the results of functional measurements (perhaps provided by expert collaborators). Without this, I view this paper as borderline in terms of providing enough new insight into MgtE channel function to justify publication in a journal with the stature of eLife.

We appreciate time and effort by the reviewer #2, as well as the positive response regarding our manuscript. As stated in the manuscript, we previously reported the [Mg^2+^]-dependent channel function measurements for the wild type and the Mg^2+^-binding mutants for the sites, 1, 2, 3, 5 and 7 (Hattori et al., 2009). The functions are summarized as follows:

Mg1-binding mutant, D432A, exhibited no channel activity.

Mg2-binding mutant, E258Q, did not close at 20 mM Mg^2+^.

Mg3-binding mutant, D259N, did not close at 20 mM Mg^2+^.

Mg5-binding mutant, D226N/D250A, did not close at 20 mM Mg^2+^.

Mg7-binding mutant, E59A, did not close at 10 mM Mg^2+^, but closed at 20 mM Mg^2+^.

Here, we have added the data for the [Mg^2+^]-dependent channel function for the rest of the binding mutants for the site 4 (D91A/D247A) and 6 (D95N), both of which have not closed at 20mM Mg^2+^ (Figure 5—figure supplement 1). These data are consistent with our NMR results that the binding mutants for the site 4 and 6 exhibited incomplete spectral changes at 4 mM Mg^2+^ (Figure 5B and Figure 5—figure supplement 3C).

We thank the reviewer #2 for the recommendation of the additional experiments. We have added expert collaborators, Drs. Tsukasa Kusakizako, Koichi Ito, and Andrés D. Maturana, to the list of co-authors.

Reviewer #3:

[…] Questions/concerns:1) Physiological range – While 0.05% of dry weight mass of E. coli is Mg^2+^, and the estimated cellular concentrations are 30 mM, the free Mg^2+^ is estimated to be around 0.3 mM for E. coli. How does the model in Figure 7 account for this range and for the estimated binding affinities?

Thank you very much for the comment on the important point. The Mg^2+^ concentration for the MgtE channel closure is proposed to be 4 mM, which is 10-fold higher than the cellular free Mg^2+^ level. Recently, we reported that ATP binding to the CBS domain of MgtE decreases the Mg^2+^-concentration for the MgtE gating (Tomita et al. Nature Comm. 2017), suggesting that the gating seems to occur at the Mg^2+^ concentration as low as 2-3 mM Mg^2+^ in the presence of ATP. However, the Mg^2+^ concentration is still much higher than the cellular free Mg^2+^ level.

Under the cellular Mg^2+^ level at 0.3 mM, the population of the Mg^2+^-bound conformation of I397^TM^, which lie in the proximity of the site Mg1, is about 0.1 (Figure 4), suggesting that MgtE exhibits Mg^2+^ conductivity at the Mg^2+^ level. When extracellular Mg^2+^ level is elevated and the excessive amount of Mg^2+^ enters into *E. coli* cells, the gate of MgtE is supposed to close.

2) It is suggested that the authors establish an explicit correlation between the behavior of the NMR-derived Mg^2+^ binding behavior at the different Mg^2+^-binding sites of MgtE, and the know Mg^2+^ dependence of the channel open probability, as determined by electrophysiological methods (perhaps as part of Figure 5?).

We are grateful for the reviewer’s suggestion. The open probability of the wild type MgtE is about 0.4, 0.2, and 0 at the Mg^2+^ concentrations of 0, 5, 20 mM Mg^2+^, respectively (Hattori et al.. 2009). In addition, the Mg^2+^ binding mutant Mg1 does not show the conductance, whereas the Mg^2+^ binding mutants, Mg2, 3, 5, and 7 do not close at 20 mM Mg^2+^. Here, we have executed patch clamp analyses for the Mg^2+^ binding mutants, Mg4 and 6, which also do not close at 20 mM Mg^2+^. The open probabilities of the wild type and mutants are plotted (Figure 5—figure supplement 1C).

As shown in Figure 5 and Figure 5—figure supplement 3, the NMR spectra for the Mg^2+^-binding mutants, Mg2, 3, and 5 in the presence of 4-5 mM Mg^2+^ exhibit high similarity to the spectrum of the wild type in the absence of Mg^2+^, rather than that of the wild type in the presence of 5 mM Mg^2+^, indicating that these mutants do not form a structure in the closed state, in the presence of 4-5 mM Mg^2+^.

Other Mg^2+^-binding mutants, Mg4, 6, and 7 in the presence of 4-5 mM Mg^2+^ provided NMR spectra, which are similar to that of the wild type protein in the presence of 5 mM Mg^2+^. However, the NMR signals from the CBS domain, particularly I190^CBS^, which lie at the interface between two CBS domains in the dimer, were not observed. This NMR data suggest that the relative orientation of the two CBS domains of these mutants are not fixed even in the presence of 4-5 mM Mg^2+^. Since the gate exists at the top of the plug helices on the CBS domains, the fluctuation of the relative orientation of the two CBS domains cannot stabilize the closed gate.

Therefore, the NMR-derived structural information on the Mg^2+^-binding mutants, Mg2-7, in the presence of saturating amount of Mg^2+^, is consistent with the electrophysiological data, in which the mutants Mg2-7 do not close in the presence of 20 mM Mg^2+^.

The differences in the Mg^2+^ concentrations for the NMR-derived conformational change (4-5 mM) and the electrophysiological experiments (20 mM) are due to the differences in the experimental conditions, such as low ion strength of 20 mM NaCl for the high sensitivity of the NMR spectra; high salt concentration, existence of the Mg^2+^-concentration gradient and membrane potential across the lipid bilayer for the patch clamp analysis.

[Editors' note: further revisions were requested prior to acceptance, as described below.]

The manuscript has been improved but there are some remaining issues that need to be addressed before acceptance, as outlined below:The main issue that the reviewers raised during the initial review was the lack of clear correspondence between the conformational responses of the protein to binding of Mg ions as detected by NMR and the functional behavior. The authors have addressed the criticism by providing functional data on mutations targeting every Mg binding site (either previously published or conducted specifically for this study). The data are succinctly summarized in the in their point-by-point response. What is still missing is putting these data in perspective of the current NMR study. In other words, how do these data agree (or disagree) with Figure 7? This is clearly a very complicated system, and perhaps not all data are expected to be in perfect correspondence, but key aspects should be addressed. For example, what is the role of sites 2, 3 and 5 if their mutations are not detrimental for channel opening but are for channel closing? I would have liked the authors to include the into the paper and discuss clearly to what extent their proposed model is or is not fully consistent with functional data.

Thank you very much for raising the important point again, which should have been appropriately addressed in the previous revision. Accordingly, we have added the figure from the previous response to the manuscript as Figure 5—figure supplement 1C.

We have addressed the correspondence between the conformational responses of the protein to binding of Mg ions as detected by NMR (summarized in Figure 7) and the functional behavior of the Mg^2+^-binding mutants shown in Figure 5—figure supplement 1 and other reported data (Hattori et al., 2009), as follows:

“Electrophysiological results indicated that a mutation at Mg1 abolishes Mg^2+^ conductivity, whereas a mutation at either of the sites Mg2, Mg3, or Mg6 does not (Hattori et al., 2009 and Figure 5—figure supplement 1C). […] The functional roles of each of these Mg^2+^-binding sites therefore underlie the intracellular Mg^2+^ homeostasis mediated by MgtE.”